# Silicon Waveguide Integrated with Germanium Photodetector for a Photonic-Integrated FBG Interrogator

**DOI:** 10.3390/nano10091683

**Published:** 2020-08-27

**Authors:** Hongqiang Li, Sai Zhang, Zhen Zhang, Shasha Zuo, Shanshan Zhang, Yaqiang Sun, Ding Zhao, Zanyun Zhang

**Affiliations:** 1Tianjin Key Laboratory of Optoelectronic Detection Technology and Systems, School of Electronics and Information Engineering, Tiangong University, Tianjin 300387, China; 1931075513@tiangong.edu.cn (S.Z.); zhangshanshan@tiangong.edu.cn (S.Z.); 1931075519@tiangong.edu.cn (Y.S.); 1732098107@tiangong.edu.cn (D.Z.); zhangzanyun@tiangong.edu.cn (Z.Z.); 2School of Computer Science and Technology, Tiangong University, Tianjin 300387, China; zhenzhang@tiangong.edu.cn; 3Tianjin Textile Fiber Inspection Institute, Tianjin 300192, China; sasa_51888@163.com; 4Tianjin Key Laboratory of Optoelectronic Sensor and Sensing Network Technology, Institute of Modern Optics, Nankai University, Tianjin 300071, China

**Keywords:** photodetector, germanium, PIN, silicon photonics

## Abstract

We report a vertically coupled germanium (Ge) waveguide detector integrated on silicon-on-insulator waveguides and an optimized device structure through the analysis of the optical field distribution and absorption efficiency of the device. The photodetector we designed is manufactured by IMEC, and the tests show that the device has good performance. This study theoretically and experimentally explains the structure of Ge PIN and the effect of the photodetector (PD) waveguide parameters on the performance of the device. Simulation and optimization of waveguide detectors with different structures are carried out. The device’s structure, quantum efficiency, spectral response, response current, changes with incident light strength, and dark current of PIN-type Ge waveguide detector are calculated. The test results show that approximately 90% of the light is absorbed by a Ge waveguide with 20 μm Ge length and 500 nm Ge thickness. The quantum efficiency of the PD can reach 90.63%. Under the reverse bias of 1 V, 2 V and 3 V, the detector’s average responsiveness in C-band reached 1.02 A/W, 1.09 A/W and 1.16 A/W and the response time is 200 ns. The dark current is only 3.7 nA at the reverse bias voltage of −1 V. The proposed silicon-based Ge PIN PD is beneficial to the integration of the detector array for photonic integrated arrayed waveguide grating (AWG)-based fiber Bragg grating (FBG) interrogators.

## 1. Introduction

Silicon (Si) photonics has been an interesting research topic in recent years due to the potential capability of monolithic integration with complementary-metal-oxide-semiconductor microelectronic circuits [1,2,3]. Photonic integrated devices are classified into different device types according to their different implementation functions, such as laser, modulator, and passive coupler [4,5,6]. We proposed an on-chip arrayed waveguide grating (AWG) interrogator by using the III-V/Si photonic integration technology. Figure 1a shows the schematic of the on-chip AWG interrogator. The optical signal enters the multimode interference (MMI) coupler through the vertical grating coupler and enters four fiber Bragg grating (FBG) distributed sensors in series through the waveguide. The FBG sensors reflect the optical signals with sensing information back to AWG, which distributes the optical signals of different wavelengths to different channels, and then converts the optical signals into electrical signals for demodulation through the array of photodetectors (PDs). Therefore, PDs play an important role in the chip. The direct epitaxial growth PDs is an optimal on-chip PD design, which has the highest optical coupling efficiency. However, most PDs are made of group III–V materials [7,8,9,10]. In addition, the lattice mismatch of group III–V and Si materials is relatively large, which is not conducive to the high-quality monolithic integration of photonic chips. The Ge material in the optical communication band has a higher absorption coefficient and a higher electron and hole mobility in the near infrared band than the group III–V semiconductor. Thus, Ge is compatible with Si microelectronic technology and has become the best choice for the Si substrate integrated PD. The waveguide-integrated PD has been widely studied because its light absorption direction is perpendicular to the carrier transport direction, and no mutual restriction exists between quantum efficiency and responsiveness [11,12,13,14,15]. Although the Ge waveguide detector has achieved progress in terms of bandwidth and responsiveness, many problems must be solved, such as the large dark current of the device and the large lattice mismatch of Si and Ge. Therefore, a type of Ge material waveguide PD is introduced to realize Si-based monolithic photonic integration. This study theoretically discusses the structure of Ge PIN and the effect of the PD waveguide parameters of the performance of the device. The research also simulates and optimizes the waveguide detectors with different structures.

## 2. Device Structure and Simulation Results

The Ge photodetector can be integrated with an Si waveguide through the end or vertical optimal coupling between the Si and Ge waveguide, thus completing the photoelectric detection in the Ge layer [16,17,18,19]. In the end-coupled structure, the active region of the Ge photodetector is parallel to the Si waveguide [20,21,22]. The integrated approach has the advantage of a short absorbing length. However, because of the limitation of the material growth process level areas, the Si and Ge waveguide coupling space interval, which can cause serious coupling loss, affects the optical signal reception. As for the vertically coupled structure, the Ge waveguide is grown directly on top of the Si waveguide, which requires no additional etching process. The integration technology is simple and can greatly improve the light absorption efficiency of the Ge detector. Figure 1b presents a vertically coupled Ge waveguide detector. The upper layer is a Ge waveguide, and the lower layer is an Si waveguide with an Si-on-insulator substrate. The proposed Ge PD can directly grow on an on-chip AWG interrogator without bonding.

The optical signal entering the detector has two incident directions. The first direction is the vertical incidence, in which the transmission direction of light is perpendicular to the plane of the PN junction and parallel to the motion direction of the photogenerated carriers. The second direction is the side incidence, in which the transmission direction of light is parallel to the plane of the PN junction and perpendicular to the motion direction of the photogenerated carriers. For the vertically incident structure, when the absorption coefficient is constant, the thickness of the absorption region must be increased to improve the quantum efficiency of the device, which can increase the carrier transit time and reduce the bandwidth. Quantum efficiency and bandwidth restrict each other. Thus, these two factors must be considered. However, for the side-incident waveguide structure, where the light travels in a parallel direction and the carrier travels in the vertical direction, quantum efficiency and carrier travel time are no longer mutually restricted. The transmission path of light can be infinitely long, while the current-carrying transit path can be as short as possible. A short transit path is beneficial to the improvement of quantum efficiency and bandwidth and is easy to integrate with other devices. In the waveguide integrated detector, the light absorption direction is the length direction of the active region of the detector, and the carrier transport direction is the thickness direction of the active region of the detector. Therefore, the length and thickness of the active region of the detector can be optimized to achieve high-performance PDs with high response speed, high quantum efficiency, and low dark current.

The performance of waveguide detectors is closely related to transmission, coupling, and absorption of light. Thus, the device should be designed according to its optical performance. We used Lumerical FDTD Solutions to build the simulation model, and analyzed the established model by Finite-Difference Time-Domain (FDTD) method. The optical absorption process of the waveguide Si–Ge detector was studied through modeling and simulation. The influence of the structural parameters of the waveguide detectors with different coupling structures on the performance of the device was is analyzed using FDTD methods. In the vertically-coupled waveguide detector, optical signal is transmitted in Si waveguide and coupled to the Ge waveguide layer. Due to the size of the Si waveguide structure, optical signal cannot be transmitted in single mode with low loss and high localization. Coupling with a Ge waveguide detector leads to a decrease in the coupling efficiency, responsivity, and sensitivity of the detector. To improve the detector performance and the coupling efficiency, the dimensions of the Si waveguide structure, which can achieve the single-mode transmission of light in a Si waveguide detector and match the propagation mode of the Ge waveguide detector, must be accurately designed. The structure without taper is illustrated in Figure 2a, which shows that the ridge width of the waveguide is 500 nm, and the length is 20 μm. The transverse optical field distribution of the simulated Si waveguide detector is illustrated as Figure 2b, and the longitudinal optical field distribution of a Si–Ge waveguide detector without a taper structure is shown in Figure 2c. The structure with taper is illustrated in Figure 2d. The transverse optical field distribution of the simulated Si waveguide detector is illustrated as Figure 2e, and the longitudinal optical field distribution of a Si–Ge waveguide detector without a taper structure is shown in Figure 2f. The results indicate that the waveguide with the taper structure at a wavelength of 1.55 μm can ensure that more than 95% of the light energy is converted to the mode energy in the wide waveguide, whereas only about 80% of the light energy in the structure without taper is converted to the mode energy in the wide waveguide as shown in Figure 3a. Therefore, the Si taper structure is selected as the input field for the detector.

In the waveguide set detector, the light absorption direction is the length direction of the active region of the PD, while the carrier transport direction is the thickness direction of the active region. The light absorption direction is independent of the width direction of the active region. The above Si taper structure is used as the input field of the detector to calculate the coupling of the Si waveguide and the detector Ge absorption layer. As shown in Figure 3b, the relation curve between Ge layer width and light absorption efficiency is obtained by simulation when the Ge layer length is 55 μm and the thickness is 600 nm. Since the Ge layer thickness is far beyond the critical thickness and strain is not introduced then strain calculation is ignored in the simulation. However, in order to minimize the impact, we doped a certain concentration of element carbon (C) between the Si and Ge layer, thus effectively improving the lattice mismatch between Si and Ge. The simulation results show that the width of germanium layer has the least effect on the light absorption efficiency.

The Ge layer has a width of 8 μm and a length of 55 μm. The percentage of optical power absorption as a function of the thickness of the Ge layer is illustrated in Figure 3c. As the thickness of the Ge layer increases, the absorption percentage of optical power gradually increases and then tends to be stable. When the thickness of the Ge layer is 500 nm, the absorption percentage of light by the Ge layer is near 90%. Coupling efficiency between the Si waveguide and the Ge absorption layer is high. Most of the light in the Si waveguide directly coupled to the Ge layer under this condition. Therefore, the optimal Ge layer thickness is 500 nm. To determine the length of the waveguide detector, the detector performance of the same device structure with Ge absorption layer areas of 10 μm × 8 μm, 15 μm × 8 μm, 20 μm × 8 μm, 25 μm × 8 μm, 30 μm × 8 μm, 35 μm × 8 μm, 45 μm × 8 μm, and 55 μm × 8 μm was calculated with a thickness of 600 nm. The results show that the Ge layer absorption efficiency increases as the Ge layer length increases. Figure 3d shows the fitting results of the absorption efficiency of the Ge layer change with the Ge layer length. The absorption efficiency of light gradually increases as the Ge layer length increases. When the thickness of the Ge layer is 600 nm, the Ge waveguide detector with a length of only 15 μm can absorb more than 80% of the light, whereas the Ge waveguide detector with a length of 20 μm can absorb more than 90% of the light. The growth rate of the light absorption percentage slows down. Thus, the Ge layer with a length of 20 μm is designed. The optical field distribution of the photodetector with a Ge layer of 55 μm is obtained by simulation. Figure 3e demonstrates the light field distribution of the Si waveguide. Figure 3f shows the optical field distribution of the Ge waveguide of the Si–Ge detector. The results indicate that the optical field is absorbed in the 20 μm length of the Ge waveguide detector, indicating that most of the light is coupled to the Ge layer. In the structure of the vertically coupled Ge waveguide detector, the thickness of the Ge layer was 500 nm, the length was 20 μm, and the width was 8 μm. The device can possess a coupling efficiency of more than 90%.

The PD structure is designed using the L-edit software, including 10 layers of optical engraving. First layer formed with taper Si waveguide, the second and third layer forming P+ doped region and set up the positive electrode heavily doped P zone formation, the fourth layer forming epitaxial Ge area and is located in the center of Si waveguide, the fifth and sixth layers will form the N+ doped region and set up the negative electrode form heavily doped N area, the seventh and eighth layer respectively to form heavy mineral P and N area to form ohm contact with metal, the ninth layer metal contact hole filling metal tungsten formation, the tenth layer form a Bond Pad metal electrodes. After adding electrodes, the overall length of the PD is 500 μm and the width is 300 μm.

To analyze the performance of the Ge waveguide detector, simulation calculations on the optoelectronic characteristics of the device are performed. The structure of the device is shown in Figure 4a, and the electric field distribution of the device is illustrated in Figure 4b. The main parameters characterizing the performance of the detector are quantum efficiency, responsivity, response speed, and dark current. The responsivity of the PD, which is used to characterize the ratio of the unit incident light power to the generated current, is defined as
(1)$$R=\left(I_{ph}\right)/\left(P_{opt}\right)=\left(q\lambda \eta \right)/\left(hc\right)$$
where *I_ph_* is the photocurrent generated by the detector, *P_opt_* is the incident light power, *q* is the electronic quantum, *λ* is the wavelength, *η* is the quantum efficiency, *h* is the Planck constant, and *c* is the light propagation speed in vacuum. Responsiveness can reflect the ability of the detector to convert the incident optical signal into an electrical signal according to Equation (Equation 1). With the increase of reverse bias, the photons absorbed in the subspace charge region of the PD increase as well. Thus, the responsivity of the PD increases. Figure 4c shows the responsivity variation as a function of wavelength. The response rate of the PD increases with the increase of reverse bias voltage.

Dark current is a key factor in investigating the performance of PDs. In general, the dark current is noise, which is very harmful to PDs, and should be minimized. The PD is a P–I–N structure, and P and N regions are heavily doped regions. The doping concentration in N-type doped regions is one order of magnitude higher than that in P-type doped regions, which can make the thickness of the N-type doped thinner so as to improve the light absorption in the active region of the detector. Therefore, the N-type doped concentration affects the dark current of the PD. In the simulation process, the P-type doped is 2 × 10^14^ cm^−3^ and the N-type doped is 2 × 10^21^ cm^−3^, 2 × 10^20^ cm^−3^, 2 × 10^19^ cm^−3^, 2 × 10^18^ cm^−3^. Figure 4d shows the change in dark current of the PD with applied bias voltage. The higher the N-type doped is, the smaller the dark current will be. When the doping concentration is more than 2 × 10^18^ cm^−3^, the increasing inhibition on the dark current will be weakened. This outcome affects the other performance of the PIN PD. The simulation results show that when the N-type doped is approximately 2 × 10^18^ cm^−3^, the dark current of the device has the best value.

## 3. Performance Analysis

The designed PD’S active area measures 20 μm × 8 μm and consists of a 2 × 10^14^ cm^−3^ P-type doped upper Si layer, a 1 × 10^15^ cm^−3^ I-type doped upper Si layer, and a 2 × 10^18^ cm^−3^ N-type doped upper Ge layer. The PD we designed were manufactured by IMEC, and the details of the PD are shown in Figure 5. The SANTEC tunable semiconductor laser 510 was used as the light source for the test. The spectral response curves of the Ge waveguide detector with different incident wavelengths are illustrated in Figure 6a. The following test results were obtained by testing the PD. The output current curve of the PD with a quantum efficiency of 100% is shown as, and the curve is shown as when the internal loss of the detector is ignored. Moreover, the cathode current of the PD is shown as, which is the actual photo response current curve. The calculated internal and external quantum efficiencies of the PD are illustrated in Figure 6b. The results show that the quantum efficiency can reach 90.63% at 1.55 μm under ideal conditions. To further verify whether the simulation with the PD structure is in line with the actual situation, the change in the photo response current of the PD with the incident optical power is calculated. The photo response current linearly varies with the incident light power, as shown in Figure 6c, indicating that the PD possesses the basic requirements. The light source is supplied to the sample to be tested, and the optical power is selected to be 6 mW. The response current of the Ge waveguide detector is shown in Figure 6d. We tested the samples using Keithley 4200 Semiconductor Characterization System. The bias setting is performed on the tested samples through the analyzer, under the reverse bias of 1 V, 2 V and 3 V, the detector’s average responsiveness in C-band reached 1.02 A/W, 1.09 A/W and 1.16 A/W. The response time reflects the detector response speed. For the PIN PD, the wider the I layer, the greater the response current but the longer the response time. The response time is 200 ns when the Ge layer width is 8 μm, as illustrated in Figure 6e, which indicates that the detector satisfies the high-speed operation. The IV characteristics under light conditions are studied. The dark current is approximately 3.7 nA at a reverse bias of −1 V, as shown in Figure 6f. The reverse saturation current is approximately 1× 10^−7^ A in the presence of light. The results indicate that the device has good rectification characteristics, and the dark current exponentially increases with the reverse bias. Testing and analysis showed that the device we designed has good performance. Table 1 compares the demonstrated device performance among different PIN PDs reported thus far in literature, and compares them with the devices developed by us.

## 4. Conclusions

The Ge waveguide detector with different structures was studied in this paper, and the detector with vertical coupling structure was optimized. More than 90% of the light can be absorbed with a length of 20 μm and a thickness of 500 nm of the detector. The quantum efficiency of the PD is 90.63%. The bias setting is performed on the tested samples through the analyzer, under the reverse bias of 1 V, 2 V and 3 V, the detector’s average responsiveness in C-band reached 1.02 A/W, 1.09 A/W and 1.16 A/W and the response time is 200 ns. Under the reverse bias of −1 V, the dark current is only 3.7 nA. This study promotes the research on the miniaturization of the fiber grating demodulation system and lays the foundation for the further development of Si-based photonic integration technology.

## Figures and Tables

**Figure 1 nanomaterials-10-01683-f001:**
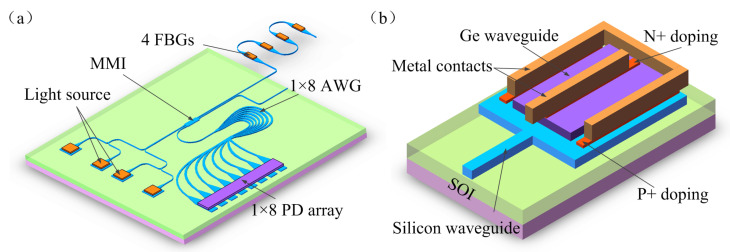
Device structure diagram. (**a**) Schematic of the on-chip arrayed waveguide grating (AWG) interrogator. (**b**) Proposed vertically coupled Ge waveguide detector.

**Figure 2 nanomaterials-10-01683-f002:**
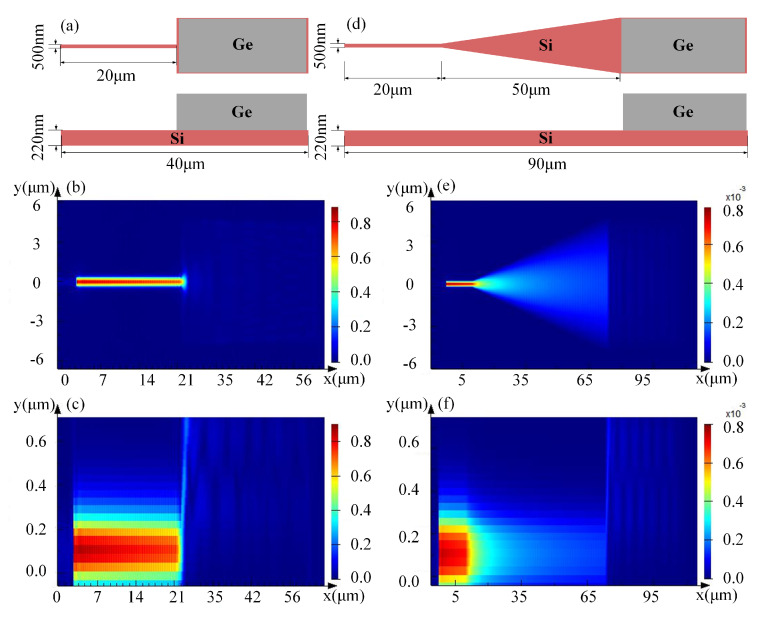
Structure and field diagram of Ge waveguide detector. (**a**) Overhead and elevation views without taper structure. (**b**) Transverse optical field distribution of Si waveguide without taper structure. (**c**) Longitudinal optical field distribution of Si–Ge waveguide without taper structure. (**d**) Overhead and elevation views of Si taper structure. (**e**) Transverse optical field distribution of Si waveguide with Si taper structure. (**f**) Longitudinal optical field distribution of Si–Ge waveguide with Si taper structure.

**Figure 3 nanomaterials-10-01683-f003:**
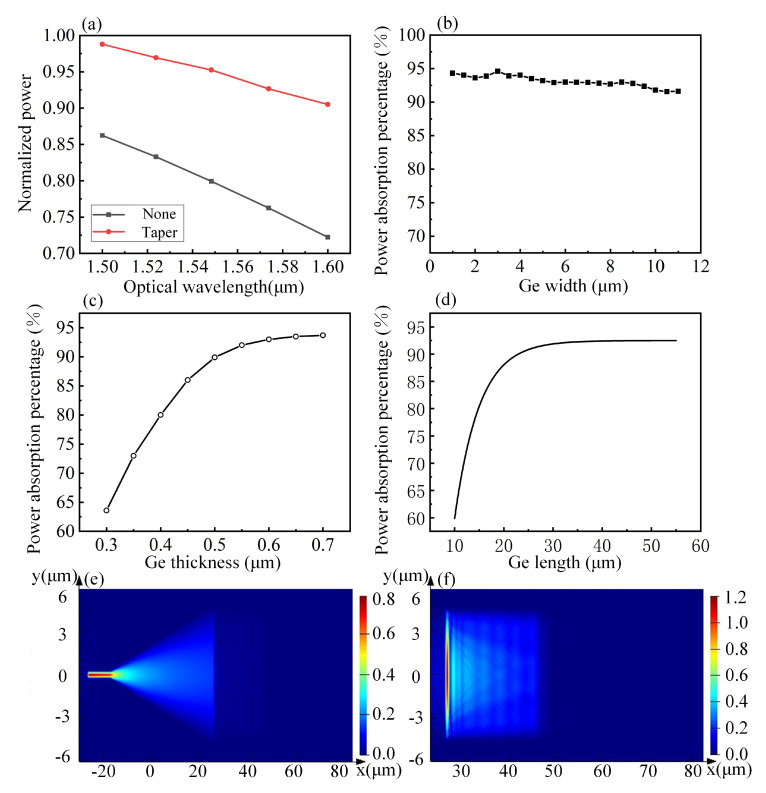
Structural simulation of Ge layer. (**a**) Normalized efficiency of energy transmission without taper structure and Si taper structure. (**b**) Variation curve of the absorption efficiency with Ge layer width. (**c**) Variation curve of the absorption efficiency with Ge layer thickness. (**d**) Variation curve of the absorption efficiency with Ge layer length. (**e**) Optical field distribution of the Si waveguide with a Si–Ge detector. (**f**) Optical field distribution of the Ge waveguide of the Si–Ge detector.

**Figure 4 nanomaterials-10-01683-f004:**
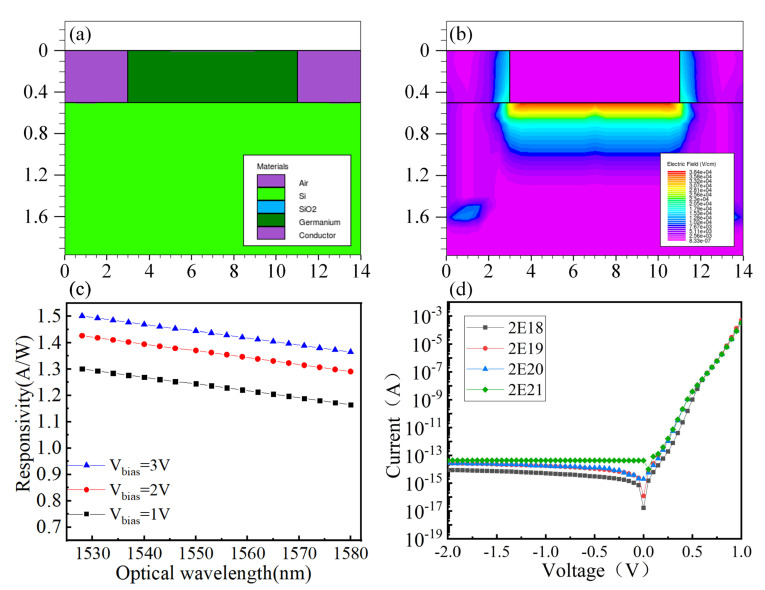
Structural performance and calculation of devices. (**a**) Structure simulation of the photodetector. (**b**) Electric field distribution of photoelectric detector. (**c**) Responsivity at different bias pressures. (**d**) Dark current at different doping concentrations.

**Figure 5 nanomaterials-10-01683-f005:**
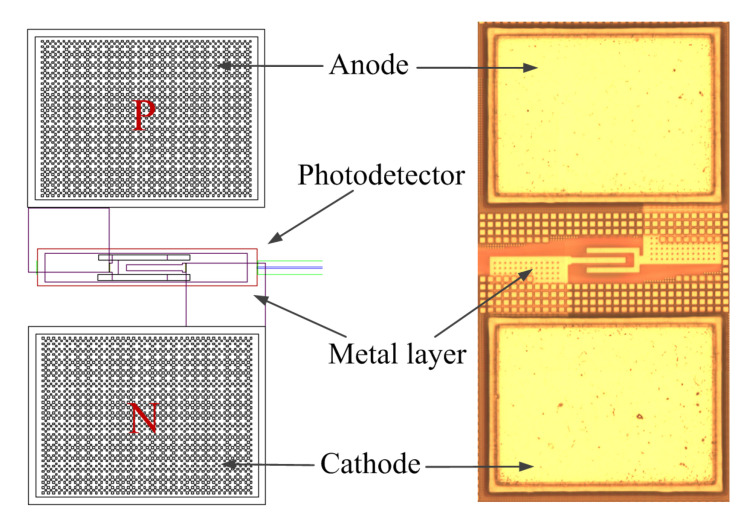
Layout of the photodetector (PD) and micrograph of the PD.

**Figure 6 nanomaterials-10-01683-f006:**
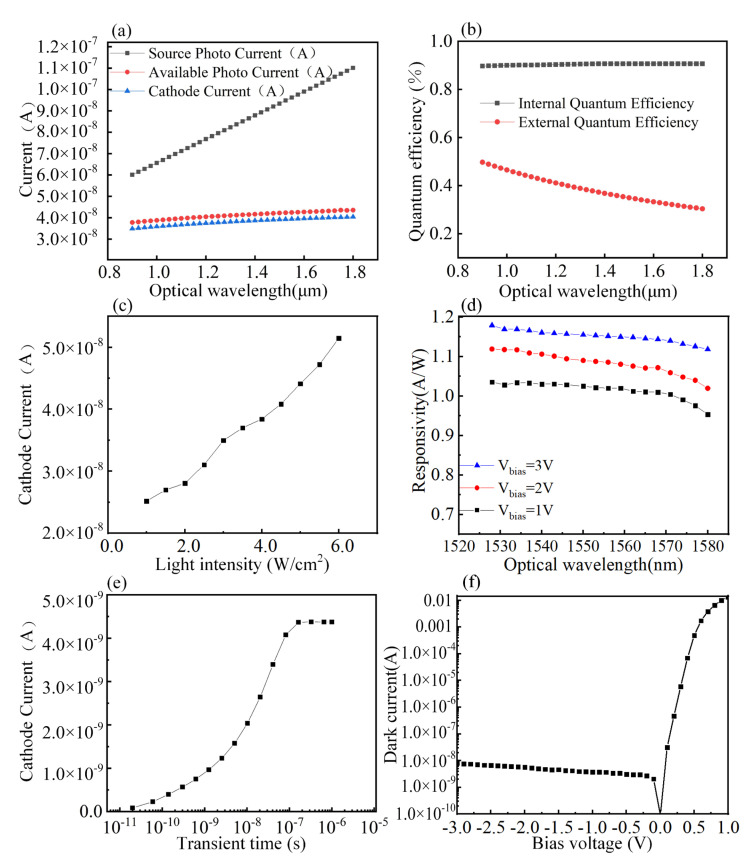
Device performance test results. (**a**) PD spectral response curve. (**b**) PD quantum efficiency curve. (**c**) Photo response current with different incident light powers. (**d**) PD response curve. (**e**) PD response time. (**f**) PD I–V characteristics.

**Table 1 nanomaterials-10-01683-t001:** Comparison of different PIN photodetector.

	Device Structure	Dark Current @-1V	Responsivity @-1V@1.55 μm
Ref. [11]	PIN	20 nA	0.56 A/W
Ref. [12]	PIN	4 nA	0.74 A/W
Ref. [13]	WG PIN	2.5 nA	0.74 A/W
Ref. [14]	PIN	11 nA	0.95 A/W
Ref. [15]	PIN	4 nA	0.92 A/W
This work	PIN	3.7 nA	1.02 A/W

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
