# Peer review of "Silicon Waveguide Integrated with Germanium Photodetector for a Photonic-Integrated FBG Interrogator"

_nanomaterials, 2020, doi:10.3390/nano10091683_

Round 1

Reviewer 1 Report

Dear editor,

I carefully reviewed the manuscript from Hongqian Li et a. entitled “Silicon Waveguide Integrated with Germanium Photodetector for Photonic-integrated FBG Interrogator”. The work seems to be technically sound, but many details about the methodology are missing, especially in the last section. Moreover, the overall English should be improved, typos need to be checked (es. Line 78) and acronyms such as FBG or AWG must be specified in the text.

Some other comments:

  • It is also not clear to me how the FBG operates, as in the figures there is no FBG.
  • How are the performances of the detector analyzed? What is the modeling tool? The authors should mention the method they used.
  • What is the light source employed in the last section for the detector performance analysis?
  • It is not clear how the authors would solve the issue of lattice mismatch between Ge and Si, which prevents the monolithic integration of the proposed devices

In conclusion, I would recommend improving the manuscript substantially. Moreover, I would consider a different journal as it seems to be rather far from the scope of Nanomaterials.

Author Response

Response to Reviewer 1 Comments

Point 1: The overall English should be improved, typos need to be checked (es. line 78) and acronyms such as FBG or AWG must be specified in the text.

Response 1: We thank the reviewer’s comment. Your comments on our manuscript have greatly improved the rigor of the paper. The errors on line 78 have been corrected. We have revised some of the figures and abbreviations in the manuscript. Figure 1 and abbreviations (PD, FBG, AWG) has been modified on Page 1, line 4 and lines 13-14 ; Page 2, Fig 1.

Point 2: Many details about the methodology are missing, especially in the last section.

Response 2: We have supplemented the relevant methods and test details. In this study, we used Lumerical FDTD Solutions to design and simulate SiGe devices. Lumerical FDTD Solutions provides a powerful algorithm module, which can quickly and efficiently calculate the established model. We build the simulation model, and analyzed the established model by Finite-Difference Time-Domain (FDTD) method. The device performance of the waveguide
Si–Ge detector is studied through modeling and simulation. At the same time, the samples we designed were manufactured by IMEC. We have tested the samples and the relevant test results are shown in the paper. The detailed description is on Page 3, lines 76-79.

Point 3: It is also not clear to me how the FBG operates, as in the figures there is no FBG.

Response 3: We have modified the model in Figure 1: Schematic of the on-chip AWG interrogator, and now you can clearly see how FBG is connected and how it works. In this test, we connected the sample to be tested through four distributed FBG in series as optical sensors. The four FBG in series with different central wavelength can realize distributed sensing. The modified system model is as follows: Fig 1. Schematic of the on-chip AWG interrogator. (Please see attachment)
“The optical signal enters the multimode interference (MMI) coupler through the vertical grating coupler and enters four FBG distributed sensors in series through the waveguide. The FBG sensors reflect the optical signals with sensing information back to AWG, which distributes the optical signals of different wavelengths to different channels, and then converts the optical signals into electrical signals for demodulation through the array of PDs.” We
elaborate the specific working mode of the system, and you can find it on Page1-2, lines 22-27.

Point 4: How are the performances of the detector analyzed? What is the modeling tool?

Response 4: We tested the samples using Keithley 4200 Semiconductor Characterization System. The bias setting is performed on the tested samples through the analyzer, we test the corresponding device response under the reverse bias of 1 V, 2 V and 3 V. We used Lumerical FDTD Solutions to build the simulation model, and analyzed the established model by FiniteDifference Time-Domain (FDTD) method. We added the specific description on page 7, lines 76-79 and lines 174-176.

Point 5: What is the light source employed in the last section for the detector performance analysis?

Response 5: “SANTEC tunable semiconductor laser 510 used as the light source for the test.” “The light source is supplied to the sample to be tested, and the optical power is selected to be 6mW.” We added the specific description on page 7, lines 161-162 and lines 170-173.

Point 6: It is not clear how the authors would solve the issue of lattice mismatch between Ge and Si, which prevents the monolithic integration of the proposed devices.

Response 6: As you mentioned, In the design and simulation we noticed that there was a 4.2% lattice mismatch between Si and Ge, but due to the Ge waveguide detector we designed has a Ge layer thickness of 600nm, which is far beyond the critical thickness, without the introduction of strain, this slight effect is often ignored in the simulation process. Moreover, the chips we designed are manufactured by IMEC, which provides an optimized process for
SiGe devices. IMEC has the process to introduce a buffer layer between Ge layer and Si layer, which is usually doped with appropriate amount of carbon (C) element, which can greatly improve the lattice mismatch between Si and Ge. And we find this is also a very effective way to deal with Si and Ge lattice mismatches. Comment on the Ge layer thickness has been added on page 3, lines 105-107.

Point 7: The manuscript requires a substantial proof-reading.

Response 7: We thank the reviewer’s comment. I has been carefully proof read and the errors have been corrected. Your comments are our wealth. We deeply appreciate your consideration  of our study, and we look forward to receiving further comments from you. We still hope that the manuscript we provided can be published at Nanomaterial. We would like to express our heartfelt thanks for your work, which will greatly facilitate our follow-up related research.

Reviewer 2 Report

Reviewer #1:
In this work, Li et al. reported a systematic study about silicon waveguide integrated with germanium photodetector. They carried-out experimental and simulation work and study quantum efficienty, spectral response, response current characteristics. Therefore, I recommend its publication, with suggested modifications, as below:
Abbrivate the FBG in text and AWG in Figure 1.
What is the power source for responsivity mesurements and how much power used in Figure 4.
If possible measure the optical spectrum for different doping concentrations (2E18, 2E19, 2E20 and 2E21) in the near infrared wavelength region.

Author Response

Response to Reviewer 2 Comments

Point 1: Abbrivate the FBG in text and AWG in Figure 1.

Response 1: We thank the reviewer’s comment. We have revised the abbreviations and this comment contributes to the rigor of the paper. We have modified the model in Figure 1: Schematic of the on-chip AWG interrogator, and now you can clearly see how FBG is connected and how it works. In this test, we connected the sample to be tested through four distributed FBG in series as optical sensors. The four FBG in series with different central wavelength can realize distributed sensing. The modified system model is as follows: Figure 1 and abbreviations (PD, FBG, AWG) has been modified on Page 1, lines 13-14. (please see attachment)

Point 2: What is the power source for responsivity mesurements and how much power used in Figure 4.

Response 2: We chose an adjustable light source because it was good for testing the sample. Equipment related models and test data have been supplemented. “ SANTEC tunable semiconductor laser 510 used as the light source for the test.” “The light source is supplied to the sample to be tested, and the optical power is selected to be 6mW.” The specific description on page 7, lines 161-162 and lines 170-173.

Point 3: If possible measure the optical spectrum for different doping concentrations (2E18, 2E19, 2E20 and 2E21) in the near infrared wavelength region.

Response 3: Your suggestion will be of great value to our subsequent research. The sample for this test is manufactured by IMEC in Belgium. We simulated the models with different doping concentrations and selected the better performance for manufacturing in combination with the actual IMEC technology, and the sample is a probe into the integrated system of photon integrated array waveguide grating (AWG) fiber bragg grating (FBG) interrogator detector array. There are more possibilities to improve device performance in the future. But we still appreciate your suggestion, which is very helpful for our research.
